# Transient Vasodilation in Mouse 4T1 Tumors after Intragastric and Intravenous Administration of Gold Nanoparticles

**DOI:** 10.3390/ijms22052361

**Published:** 2021-02-26

**Authors:** Kamil Brzoska, Małgorzata Szczygiel, Agnieszka Drzał, Martyna Sniegocka, Dominika Michalczyk-Wetula, Eva Biela, Martyna Elas, Lucyna Kapka-Skrzypczak, Hanna Lewandowska-Siwkiewicz, Krystyna Urbańska, Marcin Kruszewski

**Affiliations:** 1Centre for Radiobiology and Biological Dosimetry, Institute of Nuclear Chemistry and Technology, Dorodna 16, 03-195 Warsaw, Poland; K.Brzoska@ichtj.waw.pl (K.B.); H.Lewandowska@ichtj.waw.pl (H.L.-S.); 2Faculty of Biochemistry, Biophysics and Biotechnology, Jagiellonian University, Gronostajowa 7, 30-387 Kraków, Poland; gosia.szczygiel@uj.edu.pl (M.S.); agnieszka.drzal@uj.edu.pl (A.D.); martyna.sniegocka@gmail.com (M.S.); dominika.michalczyk@uj.edu.pl (D.M.-W.); ewka.biela@gmail.com (E.B.); martyna.elas@uj.edu.pl (M.E.); krystyna.urbanska@uj.edu.pl (K.U.); 3Department of Molecular Biology and Translational Research, Institute of Rural Health, Jaczewskiego 2, 20-090 Lublin, Poland; lucynakapka@gmail.com; 4World Institute for Family Health, Calisia University, 62-800 Kalisz, Poland

**Keywords:** gold nanoparticles, nitric oxide, vasodilation, S-nitrosothiols

## Abstract

Gold nanoparticles (AuNPs) are foreseen as a promising tool in nanomedicine, both as drug carriers and radiosensitizers. They have been also proposed as a potential anticancer drug due to the anti-angiogenic effect in tumor tissue. In this work we investigated the effect of citrate-coated AuNPs of nominal diameter 20 nm on the growth and metastatic potential of 4T1 cells originated from a mouse mammary gland tumor inoculated into the mammary fat pad of Balb/ccmdb mice. To evaluate whether AuNPs can prevent the tumor growth, one group of inoculated mice was intragastrically (i.g.) administered with 1 mg/kg of AuNPs daily from day 1 to day 14 after cancer cell implantation. To evaluate whether AuNPs can attenuate the tumor growth, the second group was intravenously (i.v.) administered with 1 or 5 mg/kg of AuNPs, twice on day 5 and day 14 after inoculation. We did not observe any anticancer activity of i.v. nor i.g. administered AuNPs, as they did not affect neither the primary tumor growth rate nor the number of lung metastases. Unexpectedly, both AuNP treatment regimens caused a marked vasodilating effect in the tumor tissue. As no change of potential angiogenic genes (*Fgf2*, *Vegfa*) nor inducible nitric oxygenase (*Nos2*) was observed, we proposed that the vasodilation was caused by AuNP-dependent decomposition of nitrosothiols and direct release of nitric oxide in the tumor tissue.

## 1. Introduction

Over the last few decades, nanomaterials have emerged as promising tools in industry and medicine. Among the known organic and inorganic nanomaterials, gold nanoparticles (AuNPs) have attracted considerable attention for their potential applications in many areas of research and industry due to ease of their synthesis and modification. Gold in its native form has long been considered inert and consequently AuNPs are often regarded as a safe, biologically compatible type of nanoparticle. As a consequence, human exposure to AuNPs has increased enormously recently due to their use in multiple areas, such as in electronics and solar cells [1,2], in catalysis [3], and in biomedical applications including radiotherapy [4], or as drug carriers [5,6]. Despite the enormous potential for AuNPs in biology and medicine, their safety is gaining more and more attention as some research has raised concerns about the toxic effects of these particles [7,8,9,10]. Toxicity of AuNPs has been shown to depend on their physicochemical properties, such as size, charge, and surface chemistry [11,12]. The majority of toxicity studies revealed that AuNPs with a diameter greater than 4–5 nm are mostly non-toxic after acute exposures. On the other hand, AuNPs smaller than 4 nm in size become catalytically active and could be cytotoxic [12].

In cancer therapy, AuNPs are usually considered as drug carriers or radiosensitizing agents. However, their direct anticancer activity was also reported. It was mainly based on reactive oxygen species formation [13] or anti-angiogenic potential [14,15,16,17]. We have previously showed that citrate-stabilized AuNPs decrease the clonogenicity of A549 lung cancer cells and HepG2 liver cancer cells in vitro. It was accompanied by changes in cell cycle distribution, gene expression, and miRNA expression [18,19]. Gold compounds have been utilized as effective therapeutic agents for the treatment of some inflammatory diseases such as rheumatoid arthritis. However, their use has become limited due to the associated high incidence of side effects. The use of AuNPs promises to reduce this undesirable side effects, but still AuNPs cannot be regarded as completely inert and some interactions with biological systems resulting in cytotoxicity must be taken into consideration.

In the present work, we analyzed the effect of bare, citrate-stabilized AuNPs on breast tumor growth in vivo. We did not observe significant changes in 4T1 tumor growth and metastatic potential in mice after intragastric and intravenous administration of AuNPs. We have, however, for the first time observed a transient vasodilating effect of AuNPs on the tumor vasculature which should be taken into account during AuNP-based nanopharmaceuticals development.

## 2. Results

### 2.1. AuNPs Have No Significant Effect on Tumor Growth and Metastatic Potential

To assess the possible impact of AuNPs on breast cancer development and metastatic potential, BALB/ccmdb mice bearing 4T1 tumors were treated intragastrically or intravenously with AuNPs. The mean tumor volume in AuNP-treated mice was similar to that in control mice (Figure 1). At the end of the experiment, the tumor volume in mice treated i.v. with 1 mg/kg b.w. AuNPs was slightly smaller than that in control mice, but the difference did not reach the level of statistical significance (Figure 1C). Similarly, the mean number of lung metastases did not differ between treated and control mice (Figure 1B,D).

### 2.2. AuNPs Transiently Increase Volume of Blood Vessels in the Tumor

Analysis of the tumor’s blood vessels volume by means of ultrasonography revealed a transient increase in the blood vessels volume after AuNP administration. In the case of intragastric administration the most significant effect was observed on day 17 and then the effect gradually decreased, but was still statistically significant on day 21 (i.e., seven days after ceasing of AuNP administration). On day 26 the tumor’s blood vessels volume returned to the level observed in control animals (Figure 2A,B).

In line, the first intravenous AuNP administration performed on day 5 after tumor implantation resulted in a significant increase in the tumor’s blood vessels volume measured 24 h later. The result was similar for both doses of AuNPs. During the next ultrasonographic (USG) measurement on day 9 the effect was no longer observed. A second dose of AuNPs was administered intravenously on day 14 and the measurement on day 17 did not show any significant differences in the blood vessels volume between the control and AuNP-treated mice (Figure 2C,D). Representative USG images are given in Appendix A. These results show that intravenous AuNP administration results in a fast increase in the tumor’s blood vessels volume, but the effect is transient and no longer detectable as soon as three days after AuNP administration.

### 2.3. Expression of Angiogenesis Related Genes and Nitric Oxide Synthase in Response to AuNP Treatment

Due to the transient nature of the observed increase of the tumor’s blood vessels volume, it was unlikely to be the result of enhanced angiogenesis. Nevertheless, we analyzed the expression of two genes crucial for angiogenesis, namely fibroblast growth factor 2 (*Fgf2*) and vascular endothelial growth factor A (*Vegfa*) in tumor samples collected at the end of the experiment, i.e., 26 days after tumor implementation. It turned out that the *Fgf2* expression in tumors from mice treated with AuNPs intravenously was more than two-fold higher than in controls (Figure 3B). This effect was absent in intragastrically treated mice. No effect of the AuNP treatment on *Vegfa* expression was observed (Figure 3).

An alternative possibility is that the observed increase in the tumor’s blood vessels volume may result from an enhanced nitric oxide (NO) production by tumor cells. To evaluate this possibility, we analyzed the expression of the *Nos2* gene encoding inducible nitric oxide synthase in tumor samples. Indeed, in the case of intravenous AuNP administration the mean *Nos2* expression in treated mice was 2.5 times higher than in control mice, but the difference was not statistically significant (Figure 3).

As the gene expression in tumor samples was measured at the end of the experiment when the effect of AuNPs on the tumor’s blood vessels volume already disappeared, it is possible that the effect on gene expression also disappeared. To check this possibility, we treated 4T1 cells in vitro with different concentrations of AuNPs for 6 h and analyzed *Nos2*, *Fgf2*, and *Vegfa* gene expressions, as well as the NO level in a cell culture medium. A small but statistically significant increase in the mRNA level was observed for all three genes under study after treatment with 80 μg/mL AuNPs (Figure 4A). However, despite the statistical significance of the increase of *Nos2* expression, its biological significance must be taken with care, since no significant change in the NO concentration in the cell culture medium was observed (Figure 4B).

## 3. Discussion

Due to the well-documented anti-inflammatory properties of gold, its salts have been used as therapeutic agents for the treatment of inflammatory diseases, such as rheumatoid arthritis. Unfortunately, the medical use of gold salts is limited by toxic side effects [20,21]. The recent development of gold nanomaterials holds great promise for improving the beneficial actions and reducing toxic properties of gold. Beside low toxicity and anti-inflammatory activity, multiple studies demonstrated anti-angiogenic effects of AuNPs, both in vitro and in vivo. Administration of AuNPs has been shown to result in a reduced vascular density in different tumor models including colorectal cancer [15], melanoma [17], liver cancer [14], and ovarian cancer [16]. Proposed mechanisms of action include a decreased expression of the vascular endothelial growth factor A (VEGFA) and fibroblast growth factor 2 (FGF2), as well as inhibition of signaling pathways initiated by these two important angiogenic factors [22]. In the present study we report that treatment of mice bearing 4T1 tumors with AuNPs results in a transient increase in volume of the blood vessels in the tumor. This effect was observed both during intragastric and intravenous AuNP administration. Understandably, the effect disappeared faster after single intravenous administration, three days after administration the effect was no longer observed, than after prolonged intragastric administration, the effect was still observed seven days after the last dose of AuNPs. During intragastric administration, the effect increased gradually with an accumulated dose of nanoparticles. Interestingly, during intravenous administration the effect is very strong as soon as 24 h after AuNP administration and possibly can be observed even earlier, but this was not analyzed during the present study. This fast and transient nature of the effect together with the above reports on anti-angiogenic activity of AuNPs make it unlikely that the increased volume of blood vessels is a result of enhanced angiogenesis and new blood vessels formation. Nevertheless, we checked the expression of *Fgf2* and *Vegfa* in tumor samples and in 4T1 cells treated with AuNPs in vitro. A significant increase in *Fgf2* expression was observed in mice treated intravenously, but not intragastrically (Figure 3). Since an increased blood vessel volume was observed after both types of treatment, the observed changes in the *Fgf2* expression seem not to be responsible for this phenomenon. Moreover, although an increased *Fgf2* and *Vegfa* expression was observed in vitro, the changes were very small and occurred only after the highest AuNP dose (Figure 4). Thus, an increased volume of blood vessels in 4T1 tumors observed in vivo is unlikely to be due to enhanced angiogenesis, however, we cannot completely exclude the possibility that some positive long-term effect of AuNPs on angiogenesis exists as suggested by the increased *Vegfa* and *Fgf2* expression.

The more likely mechanistic explanation of the observed effect is the enhanced production of nitric oxide (NO) that regulates the vascular tone and blood flow by activating soluble guanylate cyclase in the vascular smooth muscle [23]. Endogenous NO is derived largely from enzymatic reaction catalyzed by NO synthase (NOS) involving degradation of L-arginine to L-citrulline and NO in the presence of oxygen and NADPH. Three isoforms of NOS are described: endothelial NOS (eNOS or NOS3), neuronal NOS (nNOS or NOS1), and inducible NOS (iNOS or NOS2). NOS1 and NOS3 are constitutive enzymes whereas NOS2 is inducible at the level of gene transcription and expressed by different types of cells, including cancer cells, in response to different factors, such as inflammatory cytokines (TNF-α; IL-1β, IFN-γ), endotoxins, heat shock protein Hsp70, hypoxia, and oxidative stress [24]. Therefore, we measured the *Nos2* gene expression in tumor samples and 4T1 cells treated with AuNPs in vitro. We did not observe any statistically significant changes in the *Nos2* expression in tumor samples from mice treated with AuNPs compared to control animals (Figure 3). However, in the case of intravenous AuNP administration, the mean *Nos2* expression in the treated group was 2.5 times higher compared to that in the control animals. In 4T1 cells treated with AuNPs in vitro, a significant increase in the *Nos2* expression was observed only after the highest dose of AuNPs. Given that the observed increase was small (approx. 10%) it is unlikely to be biologically significant. Moreover, it did not result in an increase of NO concentration in the cell culture medium (Figure 4).

Another possible source of NO, independent of NOS activity, is the decomposition of S-nitrosothiols (RSNOs) catalyzed by AuNPs, which has been shown to take place in blood serum [25,26,27]. RSNOs are generated by the NO-dependent S-nitrosation of thiol-containing proteins and peptides. RSNOs have much the same physiological properties as NO itself, particularly vasodilation and the inhibition of platelet aggregation. They have been identified in bodily fluids, notably as S-nitrosoglutathione and S-nitrosoalbumins [28]. Because RSNOs are relatively stable in biological fluids and can release NO via reaction with transition metal ions or other reducing agents, they are envisioned as a potential transport and delivery system for NO in the organism [29]. Because of the high affinity between gold and thiols, the characteristic S−NO bond of RSNOs breaks in the presence of gold nanoparticles, releasing NO and modifying the gold nanoparticle surface with the corresponding thiol [26]. The reaction is likely to undergo in vivo, though it did not take place in our in vitro experiments, as we did not observe any changes in NO concentration following the addition of AuNPs to the cell culture (Figure 4). The reason for this is probably the very low RSNOs concentration in the cell culture medium, whereas the level of RSNO in serum was estimated to be in the micromolar range of concentrations [30]. However, more research is needed to answer the question whether RSNOs concentration in vivo in tumor tissue is high enough to trigger significant NO production and vasodilation after AuNP administration.

Our research was focused on tumor and we did not measure the blood vessels volume in other parts of the body. We are therefore unable to say if the observed effect was systemic or restricted to tumor tissue. If the abovementioned mechanism involving RSNOs is correct, then differences in RSNO concentrations in different tissues may result in a different intensity of the vasodilation. A number of studies have indicated that abnormal S-nitrosylation is implicated in cancer development and progression [31,32]. This may result in a different RSNOs concentration in tumor tissue compared to surrounding healthy tissue. Consequently, the vasodilating effect of AuNPs could be stronger in tumor than in other tissues. Obviously, it could have important consequences for AuNP-based diagnostics and therapy and more research on this topic is necessary.

To the best of our knowledge, we have shown for the first time in vivo the vasodilating potential of AuNPs administered both intragastrically and intravenously. This effect is of great importance for the development of new anti-cancer therapies based on AuNPs. In our experiments, the vasodilating effect of AuNPs did not result in significant acceleration of tumor growth or increase in its metastatic potential. However, it is likely that a longer treatment would result in a higher tumor oxygenation allowing for faster growth. On the other hand, a higher tumor oxygenation makes it more sensitive to radiotherapy. Interestingly, AuNPs are often regarded as a promising candidate for radiosensitizing agent due to the high atomic number and mass energy coefficient relative to soft tissue. This leads to an increased probability of photoelectric interaction at lower energy levels, resulting in an increased energy deposition at the target site [33]. The observed vasodilating activity potentially makes AuNPs an even more powerful radiosensitizer.

In addition, proper targeting of therapeutics to maximize efficacy and minimize side effects is one of the principia of modern medicine, and nanomedicine in particular. This is realized by different active or passive approaches, such as stimuli-responsive and active targeting or transcitosis and the enhanced permeability and retention (EPR) effect (for recent reviews see [34,35]). Active targeting, though more effective, is also less universal, as a successful target recognition is achieved by functionalization of a nanodrug with tumor targeting moieties, such as antibodies, receptor ligands, etc. On the other end, passive targeting is more universal as it does not require tumor targeting moieties, but is based on bio-physical properties of tumor tissue vessels, such as the EPR effect. The phenomenon that nanomedicines can penetrate solid tumors due to the leakage of tumor blood vessels was first described by Maeda et al. and recently summarized in [36]. Nowadays, this is a mainstay of passive anticancer nanodrug delivery. The efficiency of the EPR effect depends, among others, on nanodrug size, tumor type, characteristics of tumor vasculature, and interstitial fluid pressure. In this context, the transient vasodilation of blood vessels induced by AuNPs is an interesting feature, as it may result in an increase of the blood vessels permeability that in turn may result in an increased EPR effect. An increased EPR effect would improve the efficacy of AuNP-based nanopharmaceuticals that would improve effectiveness of the therapy.

## 4. Materials and Methods

### 4.1. Nanoparticles

Citrate-coated AuNPs of 20 nm nominal size were purchased from NanoComposix (San Diego, CA, USA). For in vivo experiments, AuNPs were diluted in ultrapure, sterile water and administered to animals at a dose of 1 mg/kg or 5 mg/kg. The in vitro and in vivo doses used in this study were chosen based on the available literature data (for a recent review see [8]). Administration was performed intragastrically or intravenously through tail vein injection. Control animals received water. During in vitro experiments, AuNPs were diluted in a cell culture medium and used in the experiments without additional processing.

### 4.2. Animals

Female BALB/ccmdb mice, 3 months of age, were purchased from the Center of Experimental Medicine, Medical University of Bialystok, Poland. Mice were kept on a standard laboratory diet (LaboFeed B from Morawski, Kcynia, Poland) with free access to drinking water in community cages, in 12 h day/night regime. Before the experiments, the animals were quarantined and acclimatized for two weeks. All procedures were accepted by the 1st Local Ethics Committee for Experiments on Animals, permission Nos 109/2016, 215/2016, 216/2016, 217/2016.

### 4.3. 4T1 Breast Tumor Cells Implantation

BALB/ccmdb mice were injected subcutaneously into the mammary fat pad with 1 × 10^5^ 4T1 cells suspended in 100 μL of PBS. The tumor growth was visible 5 days after inoculation. The tumor volume and the volume of tumor’s blood vessels were measured using the ultrasonographic imager Vevo 2100 (VisualSonics, Toronto, ON, Canada) on day 6, 9, 13, 17, 21 and 26 after tumor implantation. The tumor volume (V) was estimated on the basis of its three perpendicular diameters according to the formula: V = π/6 (a × b × c) [37].

To evaluate whether AuNPs can prevent the tumor growth, one group of inoculated mice was intragastrically administered with 1 mg/kg body mass of AuNPs daily from day 1 to day 14 after cancer cells implantation (total dose 14 mg/kg b.m.). To evaluate whether AuNPs can attenuate the tumor growth, the second group was intravenously administered twice with 1 or 5 mg/kg b.m. of AuNPs, the first dose on day 5, when the primary tumor was visible and the second at midterm of tumor growth (day 14), a total dose 2 or 10 mg/kg b.m. Control animals received water. The pilot experiment revealed that metastases begun to appear on the surface of lungs 20–25 days after tumor implantation. Over the next five days, the number of metastases increased markedly, leading to deterioration of the animal’s condition approximately 30 days after inoculation. Thus, to minimize suffering, the animals were sacrificed at day 26 and their tissues were harvested for analysis. Metastases were macroscopically observed with a binocular magnifier and counted during animal section.

### 4.4. Cell Culture

The 4T1 mouse mammary gland carcinoma cell line was purchased from the American Type Culture Collection (ATCC). Cells were cultured in RPMI medium (Gibco, Thermo Fisher Scientific, Waltham, MA, USA) supplemented with 10% fetal calf serum (Gibco, Thermo Fisher Scientific, Waltham, MA, USA). The cells were incubated in 5% CO_2_ atmosphere at 37 °C. Asynchronous cell cultures in the exponential phase of growth were used in all experiments.

### 4.5. Nitrite Determination

The conditioned medium from cells treated with AuNPs was filtered on Amicon Ultra-0.5 centrifugal filters (Merck-Millipore) at 14,000× *g* in order to remove nanoparticles. Nitrite anion content was assayed as the measure of NO contents in the preparations, according to the method of Griess [38]. Briefly, Griess reagent, freshly prepared by mixing equal volumes of 1.0% sulfanilamide in 30% acetic acid, and 0.1% *N*-(1-Naphthyl)ethylenediamine dihydrochloride in 70% acetic acid, was added to the samples (4:1 *v*/*v*). The absorbance was measured at 542 nm in the plate reader spectrophotometer Infinite M200 (Tecan, Männedorf, Switzerland). The data were referred to the calibration curve, prepared by diluting NaNO_2_ in PBS in the range of 3.13–200 mM.

### 4.6. RNA Isolation, Reverse Transcription and Real-Time PCR

The total RNA was extracted from cell pellets and tumor tissue samples using the ReliaPrep RNA Cell Miniprep System (Promega, Madison, WI, USA) and ReliaPrep RNA Tissue Miniprep System (Promega, Madison, WI, USA), respectively. RNA concentration was measured using Quantus Fluorometer (Promega, Madison, WI, USA) and QuantiFluor RNA System (Promega, Madison, WI, USA). The RNA integrity was tested by agarose gel electrophoresis. One microgram of total RNA was converted to cDNA in a 20 μL reaction volume using High Capacity cDNA Reverse Transcription Kit ( Thermo Fisher Scientific, Waltham, MA, USA) following the manufacturer’s instructions. After reaction, cDNA was diluted to 150 μL with de-ionized, nuclease-free H_2_O. Real-time PCR was performed in a 20 μL reaction mixture containing 5 μL of diluted cDNA, 4 μL of de-ionized, nuclease-free H_2_O, 10 μL of TaqMan Universal Master Mix II no UNG (Thermo Fisher Scientific, Waltham, MA, USA) and 1 μL of TaqMan Gene Expression Assay (Thermo Fisher Scientific, Waltham, MA, USA). The following TaqMan assays were used: Mm00437306_m1 (*Vegfa*), Mm01285715_m1 (*Fgf2*), Mm00440502_m1 (*Nos2*), Mm01197698_m1 (*Gusb*). All reactions were run in duplicate. PCR amplification was carried out using 7500 Real-Time PCR System (Thermo Fisher Scientific, Waltham, MA, USA) with an initial denaturation step for 10 min at 95 °C followed by 40 cycles of 95 °C for 15 s and 60 °C for 1 min. Relative gene expression was calculated using the ΔΔCt method with *Gusb* as reference control. Calculations were done using Relative Quantification Software version 2019.2.7-Q2-19-build3 (Thermo Fisher Cloud, Thermo Fisher Scientific, Waltham, MA, USA). The statistical differences were examined by Student’s *t*-test with *P* < 0.05 considered to be statistically significant.

### 4.7. Statistical Evaluation

Except for gene expression, the statistical analysis of the obtained data was performed using Statistica 7.1 software (StatSoft, Tulsa, OK, USA). Statistical significance was evaluated using Student’s *t*-test or ANOVA followed by Tukey’s post hoc test. Differences were considered statistically significant when the *p* < 0.05.

## Figures and Tables

**Figure 1 ijms-22-02361-f001:**
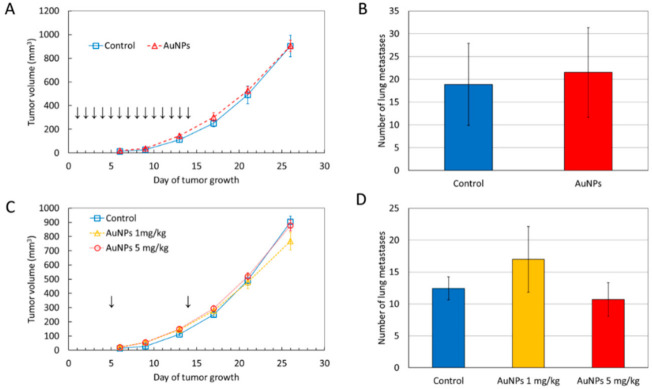
Tumor volume and the number of lung metastases in mice treated with gold nanoparticles (AuNPs) intragastrically (**A**,**B**) and intravenously (**C**,**D**). Arrows indicate the day of AuNP administration. Data are presented as the mean ± standard error from 6 mice. The differences are not statistically significant.

**Figure 2 ijms-22-02361-f002:**
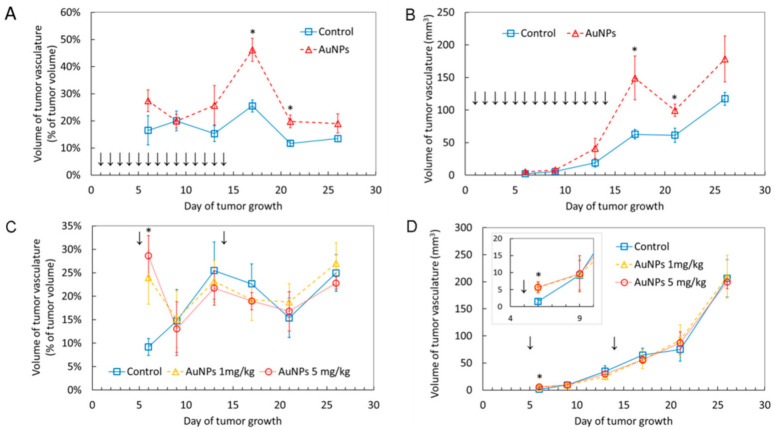
Volume of tumor’s blood vessels shown as an absolute value or as a percent of tumor volume in mice treated with AuNPs intragastrically (**A**,**B**) and intravenously (**C**,**D**). Arrows indicate AuNP administration. Data are presented as the mean ± standard error from 6 mice. * *p* < 0.05.

**Figure 3 ijms-22-02361-f003:**
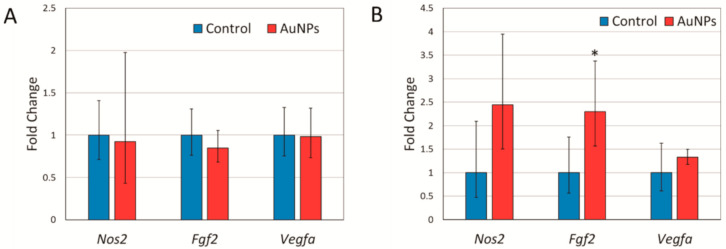
Expression of *Nos2*, *Fgf2,* and *Vegfa* at mRNA level in tumors from mice treated with AuNPs intragastrically (**A**) and intravenously (**B**). Data are expressed as means and 95% confidence intervals from six animals. * *p* < 0.05 difference versus control group.

**Figure 4 ijms-22-02361-f004:**
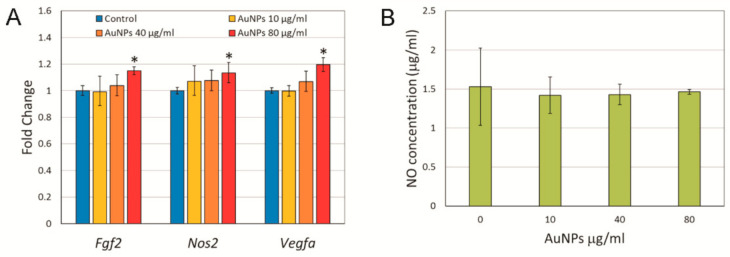
Cellular expression of *Fgf2, Nos2* and *Vegfa* at mRNA level (**A**) and NO concentration (**B**) in cell culture medium after treatment of 4T1 cells with AuNPs in vitro for 6 h. Data are expressed as means and 95% confidence intervals from four independent experiments. * *p* < 0.05 difference versus control group.

## Data Availability

The raw data are available upon request from the corresponding author.

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
