# Peer review of "Transient Vasodilation in Mouse 4T1 Tumors after Intragastric and Intravenous Administration of Gold Nanoparticles"

_ijms, 2021, doi:10.3390/ijms22052361_

Round 1

Reviewer 1 Report

The manuscript of Brzoska et al. describes the effect of intragastric and intravenous administration of Au NPs in 4T1 tumor bearing mice. 

The results display no anticancer activity and no effects on tumor growth and metastases. The treatment causes vasodilatation in tumor tissue. This represents an interesting information to take into account for future study involving AuNPs.

In my opinion it could be necessary to elucidate the choice of AuNPs concentrations and time for administration, eventually citing previous studies.

Also time and concentration used for in vitro cell treatment should be motivate.

Please also add "C" and "D" in the fig 2 and revise line 172.

Author Response

Comments and Suggestions for Authors

The manuscript of Brzoska et al. describes the effect of intragastric and intravenous administration of Au NPs in 4T1 tumor bearing mice. 

The results display no anticancer activity and no effects on tumor growth and metastases. The treatment causes vasodilatation in tumor tissue. This represents an interesting information to take into account for future study involving AuNPs.

In my opinion it could be necessary to elucidate the choice of AuNPs concentrations and time for administration, eventually citing previous studies.

Since we do not have our own experience with AuNPs, we decided to choose the in vivo AuNPs concentrations based on the available literature data. The recent review by Adewale et al. (Int. J. Tox. 2019,  38(5), 357-384, DOI: 10.1177/1091581819863130) summarizes available data on in vivo AuNPs toxicity. Despite large divergence in experimental approach, it is clear from the review that in in vivo mouse models doses up to 10 mg/kg b.w were used so far, usually up to 5 mg/kg b.w. per a single dose. This is in a perfect agreement with the doses used in this study total 2 or 10 mg/kg b.w. (1 or 5 mg/kg b.w. per a single dose). For oral administration there is only one publication available so far (Zhang et al. Int J Nanomedicine 2010;5:771-781) were doses up to 2.2 mg/kg b.w. per day (0.1375, 0.275, 0.550 1.100 and 2.2 mg/kg b.w. per day) were used for 14-day treatment (Total highest dose 30.8 mg/kg b.w.) and 1.1 mg/kg b.w. per day were used for 28-day treatment (Total dose 30.8 mg/kg b.w.). This is again in perfect agreement with the dosage used in this study (1 mg/kg b.w. per day, 14-day treatment (Total dose 14 mg/kg b.w.).

Also time and concentration used for in vitro cell treatment should be motivate.

The same review (DOI: 10.1177/1091581819863130) summarizes publications on the in vitro toxicity of AuNPs. While the dosage depends on cellular model used in the particular study, in general the doses ranged between 0 and 250 µg/mL. Though there is no publication on AuNP toxicity on 4T1 cells, the doses used in our study (10, 40 and 80 µg/mL) fit very well a general tendency.

To justify our selection of doses additional sentence was added to the Materials and methods section

“The in vitro and in vivo doses used this study were chosen based on the available literature data (for a recent review see [8]).”

Please also add "C" and "D" in the fig 2 and revise line 172.

Fig 2 and line 172 were revised. Thank you vigilance.

Reviewer 2 Report

Regarding the Brzoska et al. ijms-1102369, it indeed represents an informative and intriguing research toward tumor vascular permeability enhancement by using gold nanoparticles. It is very suitable for IJMS. The text, which lacks a bit of fluency, could be edited for clarity, though none of my further comments will disqualify its publication:

(1) In fact, improving tumor targeting is a critical mission of nanomedicine-based drug delivery. It is important to develop effective chemo-physical strategies to enhance tumor targeting, such as increasing vascular permeability using gold nanoparticle in this manuscript. The authors are encouraged to include more discussion on this point and cite some recent research progress (e.g., J. Am. Chem. Soc. 2021, 143, 2, 538–559).

(2) If possible, the authors should provide the proof of concept study for tumor vascular permeability enhancement by using gold nanoparticles, i.e., providing definitive evidence for real application, such as, enhancing the accumulation of therapeutic nanomedicine (e.g., Doxil®). No much result is needed but should prove it.

(3) Why the authors do not evaluate the normal tissue? It is very nice to prove that vascular permeability enhancement is tumor-specific.

Author Response

Comments and Suggestions for Authors

Regarding the Brzoska et al. ijms-1102369, it indeed represents an informative and intriguing research toward tumor vascular permeability enhancement by using gold nanoparticles. It is very suitable for IJMS. The text, which lacks a bit of fluency, could be edited for clarity, though none of my further comments will disqualify its publication:

We went carefully through the paper trying to improve its clarity.

(1) In fact, improving tumor targeting is a critical mission of nanomedicine-based drug delivery. It is important to develop effective chemo-physical strategies to enhance tumor targeting, such as increasing vascular permeability using gold nanoparticle in this manuscript. The authors are encouraged to include more discussion on this point and cite some recent research progress (e.g., J. Am. Chem. Soc. 2021, 143, 2, 538–559).

Though discussing the EPR effect is not the main stream of this paper, we changed the last paragraph of Discussion, to give the readers the better idea about the usefulness and potential applications of the described observation.

Current version of the last paragraph of discussion section.

“In addition, proper targeting of therapeutics to maximize efficacy and minimize side effects is one of the principia of a modern medicine, and a nanomedicine in particular. This is realized by different active or passive approaches, such as stimuli-responsive and active targeting or transcitosis and enhanced permeability and retention (EPR) effect (for the recent reviews see [34, 35]). Active targeting, though more effective is also less universal, as an successful target recognition is achieved by functionalization of a nanodrug with tumor targeting moieties, such as antibodies or its part, receptor ligands etc. On the other end, passive targeting is more universal as it does not require tumor targeting moieties, but is based on bio-physical properties of tumor tissue vessels, such as the EPR effect. The phenomenon that nanomecidines can penetrate the solid tumors due to the leakage of tumor blood vessels was first described by Maeda et al. and recently summarized in [36]. Nowadays this is a mainstay of passive anticancer nanodrug delivery. The efficiency of EPR effect depends, among others, on nanodrug size, tumor type, characteristics of tumor vasculature, interstitial fluid pressure. In this context transient vasodilation of blood vessel induced by AuNPs is an interesting feature, as it may result in increase of blood vessels permeability that in turn may result in increased EPR effect. Increased EPR effect would improve the efficacy of AuNPs-based nanophamaceuticals that would improve effectiveness of the therapy”

(2) If possible, the authors should provide the proof of concept study for tumor vascular permeability enhancement by using gold nanoparticles, i.e., providing definitive evidence for real application, such as, enhancing the accumulation of therapeutic nanomedicine (e.g., Doxil®). No much result is needed but should prove it.

We absolutely agree that any evidence of enhanced accumulation of therapeutic nanomedicine would add tremendously to this study. However, this study was a part of bigger research on the effect of AgNPs on tumor development and metastasis. Paper in preparation. Frankly, AuNPs were used as a non-toxic equivalent of 20 nm AgNPs. The vasodilating effect was a big surprise for us, as we rather expect anti-angiogenic effect and inhibition of tumor growth as described in the literature. Thus no extended study was planned. Unfortunately to prove the enhancing accumulation of therapeutic nanomedicine, a separate set of in vivo experiments should be performed. This type of experiments is labour consuming and expensive, but most of all it  requires new approval of Ethical Commission, which in our country may takes months. As the project has finished last year, it is not possible to do any additional experiments due to the money shortage. BTW AgNPs did not show any vasodilating effect.

(3) Why the authors do not evaluate the normal tissue? It is very nice to prove that vascular permeability enhancement is tumor-specific.

As mentioned above this study was a part of bigger research on the effect of AgNPs on tumor development and metastasis. Since, we did not expect any vasodilating effect of AgNP or AuNP, no investigation of normal tissue was planned. To avoid an experimental bias the whole set of data was analysed after data collection, thus any modifications of experimental design could not be made. Again we agree that any proof that the effect is tumor-specific would be nice. However, it is not possible at this stage. See reasons above.

As both nitrosoglutathione and nitrosyl iron complexes are ubiquitous carriers of NO, personally, I think that the effect is rather common for both normal and tumor tissue, however the extent of the effect can be different, as there are differences in distribution of nitrosyl iron complexes in normal and tumor tissue (see Bastian et al., JBC 1994, 269(7), 5127-31) and the literature cited therein)